# High prevalence of an alpha variant lineage with a premature stop codon in ORF7a in Iraq, winter 2020–2021

**Nihad A. M. Al-Rashedi**[1]*, **Hussein Alburkat**[2], **Abas O. Hadi**[3,4], **Murad G. Munahi**[5], **Ali Jasim**[4], **Alaa Hameed**[3], **Basel Saber Oda**[4], **Kareem Moamin Lilo**[6], **Laith A. H. AlObaidi**[1], **Olli Vapalahti**[2], **Tarja Sironen**[2], **Teemu Smura**[2]

**1** Department of Biology, College of Science, Al-Muthanna University, Samawah, Iraq, **2** Department of Virology, Faculty of Medicine, University of Helsinki, Helsinki, Finland, **3** Department of Health Administration, College of Health & Medical Technology, Sawa University, Samawah, Iraq, **4** Department of Public Health, Al-Muthanna Health Directorate, Samawah, Iraq, **5** Department of Chemistry, College of Science, Al-Muthanna University, Samawah, Iraq, **6** Directorate of Medical Affairs, Ministry of Health, Baghdad, Iraq

* nhidaee@mu.edu.iq

## Abstract

Since the first reported case of coronavirus disease 2019 (COVID-19) in China, SARS-CoV-2 has been spreading worldwide. Genomic surveillance of SARS-CoV-2 has had a critical role in tracking the emergence, introduction, and spread of new variants, which may affect transmissibility, pathogenicity, and escape from infection or vaccine-induced immunity. As anticipated, the rapid increase in COVID-19 infections in Iraq in February 2021 is due to the introduction of variants of concern during the second wave of the COVID-19 pandemic. To understand the molecular epidemiology of SARS-CoV-2 during the second wave in Iraq (2021), we sequenced 76 complete SARS-CoV-2 genomes using NGS technology and identified genomic mutations and proportions of circulating variants among these. Also, we performed an in silico study to predict the effect of the truncation of NS7a protein (ORF7a) on its function. We detected nine different lineages of SARS-CoV-2. The B.1.1.7 lineage was predominant (80.20%) from February to May 2021, while only one B.1.351 strain was detected. Interestingly, the phylogenetic analysis showed that multiple strains of the B.1.1.7 lineage clustered closely with those from European countries. A notable frequency (43.33%) of stop codon mutation (NS7a Q62stop) was detected among the B.1.1.7 lineage sequences. In silico analysis of NS7a with Q62stop found that this stop codon had no considerable effect on the function of NS7a. This work provides molecular epidemiological insights into the spread variants of SARS-CoV-2 in Iraq, which are most likely imported from Europe.

## Introduction

In late December 2019, an outbreak of pneumonia of unknown etiology was announced in Wuhan, China. A relative unknown coronavirus named "Severe Acute Respiratory Syndrome

**Data Availability Statement:** All relevant data are within the paper and its Supporting information files. Also, all raw sequencing data used in this study are available on the Sequence Read Archive

(SRA) under the BioProject accession numbers PRJNA731979, PRJNA735311, and PRJNA738286. The genome sequences were deposited in GISAID and GenBank and are now accessible by the numbers listed in S1 and S3 Tables.

**Funding:** Funding: This study was supported by Sawa University, Samawa, Iraq (fund No. SA001/2921). Alaa Hameed and Abas Hadi from the funder institution have contributed to samples and data collection and preparation of the manuscript.

**Competing interests:** The authors have declared that no competing interests exist.

Coronavirus 2 (SARS-CoV-2)" was then identified as the causative agent of COVID-19 [1]. On March 11, 2020, the WHO declared the COVID-19 outbreak a pandemic [2], affecting humans worldwide [3]. SARS-CoV-2 is highly infectious and has caused over 234 million confirmed cases of COVID-19 globally, including over 4.8 million deaths reported by the WHO as of October 3, 2021.

In Iraq, the first SARS-CoV-2 case was diagnosed in February 24, 2020 [4]. Since then, over 695,489 cases and more than 13,000 deaths were confirmed by the end of February 2021 [5]. The epidemiological situation displayed a slight improvement at the end of the first wave (week 1, 2021). However, the number of cases has risen with the beginning of the second wave in week 5, 2021 [6].

Emerging RNA viruses are a global health concern due to their potentially high transmission rate, high mutation rates, and aggressive competition to host cellular functions. As a result of SARS-CoV-2 mutation dynamics, several variants of concern have emerged, of which there is evidence of increased transmission, a surge in hospitalization, critical care unit admissions, and fatalities to varying degrees compared with the wild-type Wuhan-1 strain [7, 8], and/or evidence of decreased neutralization by antibodies raised against previous infection or vaccine [9]. In particular, amino acid replacements in the spike protein can lead to enhanced binding with the host ACE2 receptor causing increased transmissibility and potentially higher virulence [5, 10]. Similar to influenza A virus and, to a lesser extent, seasonal human coronaviruses, SARS-CoV-2 can be expected to accumulate adaptive amino acid replacements in its glycoprotein, resulting in antigenic drift [11]. However, due to the biological differences in influenza A virus (IAV) (which is a segmented negative-stranded RNA virus with a higher overall evolutionary rate compared to coronaviruses) and seasonal coronaviruses, (which have been circulating among the human population for a long time), genomic surveillance of SARS-CoV-2 is needed to detect and assess the effect of such mutations [11]. The global effort for SARS-CoV-2 sequencing has led to efficient tracking of circulating lineages as well as tracking of mutations that may lead to changes in vaccine efficacy, PCR detection, and virus transmissibility [12, 13]. Therefore, surveying the molecular epidemiology/spatiotemporal changes in the SARS-CoV-2 genome and understanding its mutations are important. Yet, there is a significant underrepresentation of SARS-CoV-2 sequences from middle- and low-income countries in the global dataset [14].

Recently, four variants have been identified by the WHO to be of particular concern (VOCs): Alpha variant (B.1.1.7), first reported in the UK; Beta (B.1.351), first reported in South Africa in October 2020; P.1, a descendant of variant Gamma (B.1.1.28), first reported in Brazil, and Delta (B.1.617.2), first reported in India [15]. Only a limited number of SARS-CoV-2 sequences are currently available from Iraq (https://www.gisaid.org/). The first sequence of Iraqi patients available from the first wave showed the presence of a GH clade with the D614G mutation [16]. In the current study, we sequenced 76 SARS-CoV-2 genomes to produce baseline data for the genomic surveillance of SARS-CoV-2 in Iraq. Our work summarizes sequences, emerging mutations, and the evolutionary relationships of SARS-CoV-2 in Iraq between December 2020 and February 2021.

## Materials and methods

### Sampling

Combined naso- and oropharyngeal swabs were collected from 76 patients (46 males and 30 females, age ranging between 13 and 85 years) in Samawa, Iraq (31.3188˚ N, 45.2806˚ E) during the second epidemic wave of COVID-19 in Iraq (between December 27, 2020 and February 28, 2021). Of these patients, five (6.6%) died, eight (10.5%) had severe disease, and the

remaining 63 (82.9%) had mild to moderate infections. The samples were analyzed with STAT-NAT COVID-19 MULTI real-time PCR kits (Sentinel, Milano, Italy), based on two targets in RdRP and Orf1b genes, to detect the presence of SARS-CoV-2. The real-time PCR assay was conducted using the Mx3000P qPCR system (Agilent Technologies, Waldbronn, Germany). A total of 76 samples that had a high copy number of the virus (Ct values <24) were selected for whole genome sequencing.

The study was conducted according to the guidelines of the declaration of Helsinki and was approved by the scientific research ethics committee of Al Muthanna University within the collaborative protocol of joint work between the College of Science and Public Health Department, Al-Muthanna Directorate (July 30, 2020–8928). All participants provided written informed consent and agreed to use their medical records for research purposes.

## SARS-CoV-2 sequencing

RNA isolation was carried out using TRIzol reagent (ThermoFisher Scientific, MA, USA) from a viral transport media (VTM) sample (3:1 ratio) according to the manufacturer's procedure. The LunaScript RT Super Mix Kit (New England Biolabs, UK) was used for first-strand cDNA synthesis. A multiplex PCR approach following the ARTIC protocol was used to amplify the viral genome using Q5 High Fidelity DNA Polymerase (New England BioLabs, UK). The NEBNext Ultra II library prep kit was used for Illumina sequencing library preparation. The libraries were quantified using the Qubit 4 with the dsDNA High Sensitivity Kit (ThermoFisher Scientific, MA, USA).

High throughput sequencing was performed using the Illumina NovaSeq 6000 system with a read length of 250 bp, which produced a range of 1.3 to 3.3 million paired-end sequence reads per sample. Additionally, sequencing reads with low-quality (quality score <30) and short sequence (<50 nt) were removed using Trimmomatic [17], assembled using BWA-MEM [18], variant called using LoFreq [19], and consensus called using SAMtools [20], implemented in the HaVoC pipeline [21].

## Genome sequence analysis

Mutation analysis of the SARS-CoV-2 genome was interpreted using the GISAID CoVsurver "CoVsurver enabled by GISAID" [22] and Coronapp web application [23]. Lineage and clade assignment were identified using Pangolin (version v.3.1.7) [24] and the Nextstrain web server [25].

## Homology structure of mutant ORF7a and molecular docking

The 3D model of the mutant ORF7a was built using the Swiss-model web server, and the crystal structure of ORF7a (pdb:7ci3) was used as a template. Structural comparison was performed between the selected template and the built model to assign their similarity and dissimilarity using TM-align [26] and FATCAT web tools [27]. The PROCHECK web server was used to validate the best-fit model based on the stereochemical properties and geometry of the structure [28]. The quality of the model was evaluated by establishing a plot between phi and psi of the polypeptide residues using the Ramachandran plot server [29]. Subsequently, the model structure was refined using a 3D refine web server [30]. Finally, the refined model structure was prepared for docking by adding polar hydrogen atoms and Gasteiger charges using the Autodock tool 1.5.6 [31]. The model, along with the wild ORF7a, was subjected to the HADDOCK 2.4 web server to investigate the protein-protein interactions [32].

### Phylogenetic tree

Genome sequence alignment was performed using alignment of multiple complete SARS-CoV-2 genomes (MAFFT online version April 11, 2020) [33]. To analyze the SARS-CoV-2 genome samples derived from the Iraqi patients in a phylogenetic relatedness, a data set of 146 available SARS-CoV-2 complete genomes from different countries was collected from GISAID available on May 25, 2021 (S1 Table). The phylogenetic tree was mapped by a maximum likeli-hood estimation using a fit substitution model (ModelFinder) and replicate number with 1000 bootstrap on IQ-TREE (version: 1.6.10) [34] with ultrafast bootstrap support. ITOL v6 tools [35] were used for the visualization of the phylogenetic tree.

## Results

The COVID-19 pandemic caused by SARS-CoV-2 has caused significant morbidity and mor-tality worldwide. During the first wave in Iraq, (February–December 2020), implementation of the restrictions (lockdowns) was related to a significant reduction in daily reported cases and mortality, followed by phased relaxation in restrictions. During the second wave (Fig 1), the cumulative number of cases reached more than a million by late May 2021 [6].

### Genetic clades and lineages of SARS-CoV-2

We identified nine different genetic lineages, including two variants of concern: B.1.1.7 (Alpha variant) and B.1.351 (Beta variant), in Iraq. Among the sequenced samples, alpha variant B.1.1.7 (n = 61, 80.2%) was the most prevalent lineage, whereas only one B.1.351 was detected (n = 1, 1.3%). Seven other SARS-CoV-2 lineages were detected: B (1.3%), B.1.1 (7.8%), B.1.177.21 (1.3%), B1.36 (1.3%), B.1.621.1 (1.3%), B.1.1.374 (1.3%), and B.1.438 (3.9%) (Fig 2).

The first designation of the B.1 lineage was reported in Iraq on June 30, 2020 [16]. Our results suggest turnover of circulating lineages, resulting as the dominance of the Alpha variant (B.1.1.7) during the last three months of sampling (December–February 2021).

### Genomic mutation analysis

Analysis of the amino acid changes between SARS-CoV-2 strains in this study and the refer-ence genome of virus EPI_ISL_402124 (hCoV-19/Wuhan/WIV04/2019) identified numerous

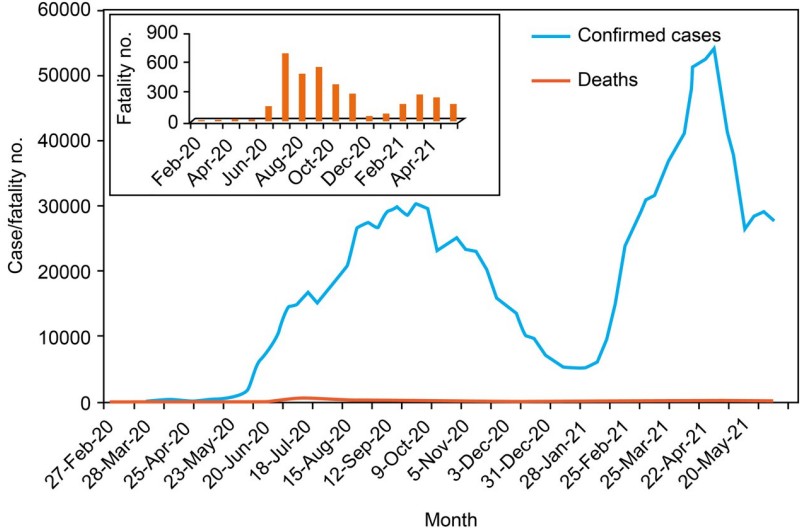

**Fig 1. COVID-19 cases/week and fatalities/week rates during first and second waves in Iraq [6].**

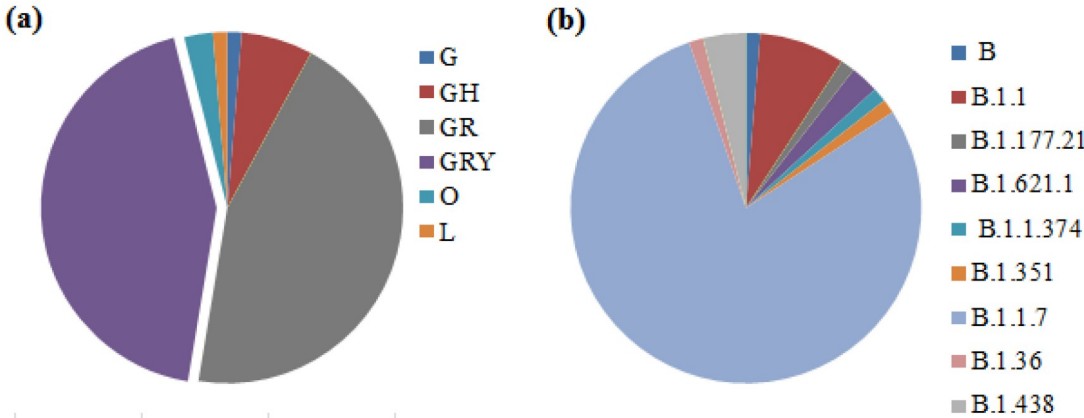

**Fig 2. Distribution of GISAID clades (a) and Pango lineages (b) of the Iraqi sequenced strains.** Lineages B.1.1.7 and genetic clades GR and GRY show the most prevalent in SARS-CoV-2 strains.

mutations detected in the 5'UTR, NSP2, NSP3, NSP4, NSP6, NSP12b, NSP13, NSP15, NSP16, Spike, ORF6, ORF7, ORF8, 3'UTR, and N.

**Multiple spike protein mutations.** The most common cluster of spike protein mutations were H69del, V70del, Y144del, A570D, D614G, P681H, T716I, S982A, and D1118H, which were detected in both GRY (n = 33, 43.4%) and GR (n = 34, 38.1%) (S2 Table).

The most notable strain in this study, one out of 76, was identified as a Beta variant, B.1.351 lineage with spike mutation profile: D80A, D215G, L242del, A243del, L244del, E484K, N501Y, D614G, and A701V, which are linked to the South African variant (20H/501Y.V2). In addition, spike mutations A222V and L18F were found in the Iraqi strain of clade O (n = 1, 1.3%). Using the web application Coronapp [22], the amino acid change D614G in the spike glyco-protein was detected at a high frequency of sequences (n = 73, 96%) (Fig 3).

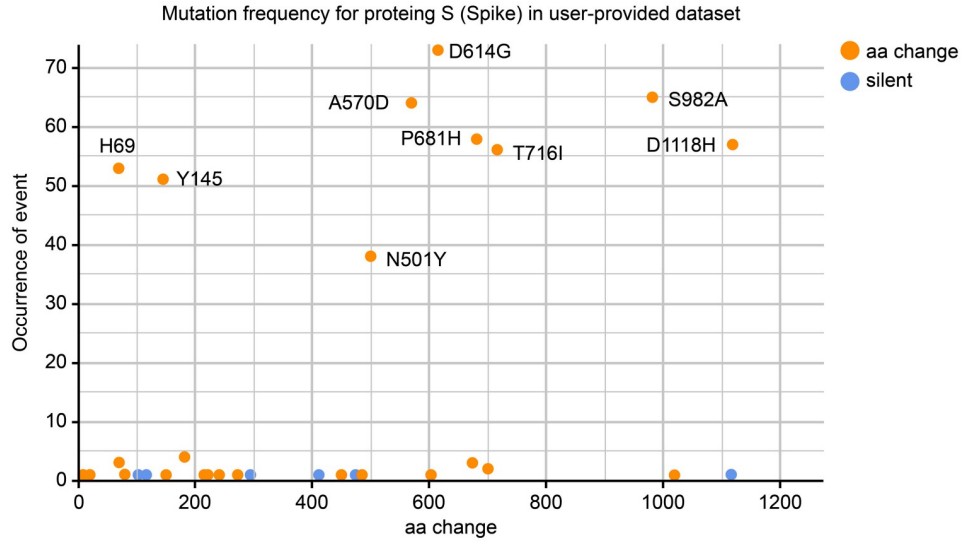

**Fig 3. Frequency of spike mutations in 76 SARS-CoV-2 genomes provided datasets.** D614G with the most frequent mutation appears in 73 of the original spike proteins data set. Red-dots indicate amino acid replacement mutations and blue-dots indicate silent mutations.

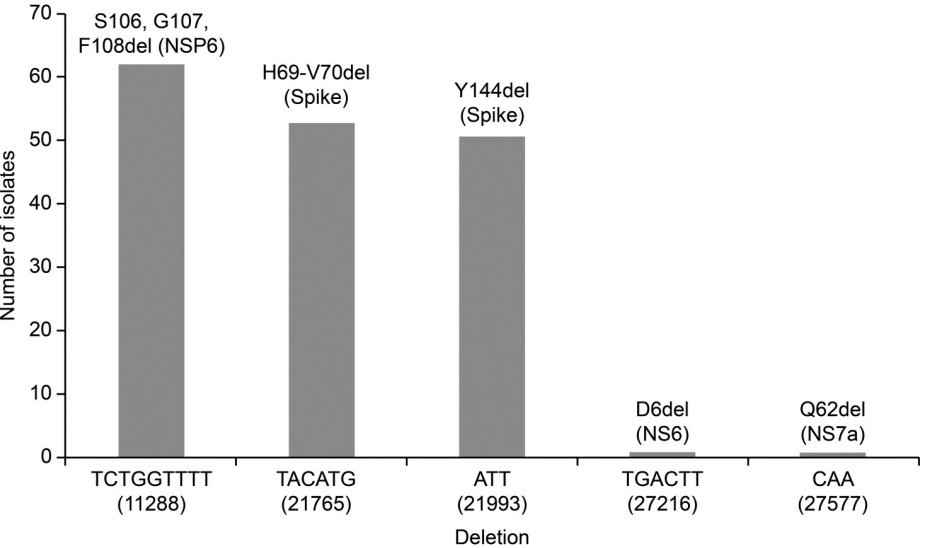

**Fig 4. Deletion mutations recorded in 67 of genomic SARS-CoV-2 sequences.**

In addition, a rare amino acid change S605A, which is located in the S1 subunit of the spike protein, was detected in one of the sequenced viruses (EPI_ISL_1524356) According to GISAID data (accessed in May 2021), this amino acid substitution has been reported in three strains, one from Iraq in this study and two from India and the USA.

**Novel aa changes.**   Among the detected mutations, four novel aa changes were identified (E45V/NSP3, M41K/NSP16, W128M/NS3, and V256D/NS3). In this study, the collected strains that individually carried these amino acid changes were EPI_ISL_2467922, EPI_ISL_2467921, EPI_ISL_1524366, and EPI_ISL_2467913, respectively.

**NS7a Q62stop.**   Amino acid change NS7a Q62stop occurred in 26 strains sequenced in this study (S3 Table) and recorded in 77 countries before.

**Deletion mutations.**   Are shown in Fig 4, comprising high frequencies of S106, G107, and F108del in NSP6 (n = 62, 81.6%), spike H69del (n = 53, 69.7%), and Y145del (n = 51, 67.1%).

## Molecular docking of ORF7a stop mutation

We found a high prevalence of premature stop codon Q62stop among the Iraqi Alpha variant strains. SARS-CoV-2 ORF7a is a transmembrane protein (type I) composed of 122 amino acids (15, 81, 21, and 5 amino acids composed to N-terminal, luminal domain, transmembrane segment, and cytoplasmic tail, respectively). It has been reported [36] that this accessory protein modulates the immune response of the host by binding with the host lymphocyte function-associated antigen I (LFA-1). Previous studies have suggested that the amino acids T39, E41, N43, Q62, A66, and K72 play a key role in the function of ORF7a [36]. Among these six active residues, two (A66 and K72) were truncated by a premature stop codon. The truncated amino acids are mainly located in one β sheet strand (βG) in the ectodomain. Only four amino acids are located in the β sheet strand βf (Fig 5). To predict the potential effect of the stop codon (Q62*) on the function of the ORF7a protein, we constructed a 3D model of the mutant ORF7a. The crystal structure of ORF7a (PDB: 7ci3) was used as the template. The crystal structure (PDB: 3f78) of LFA-1 was also used in the molecular docking simulation. The similarity and dissimilarity between the selected template and model were performed using TM-align and FATCAT, where the optimal structural similarity was evaluated based on the obtained

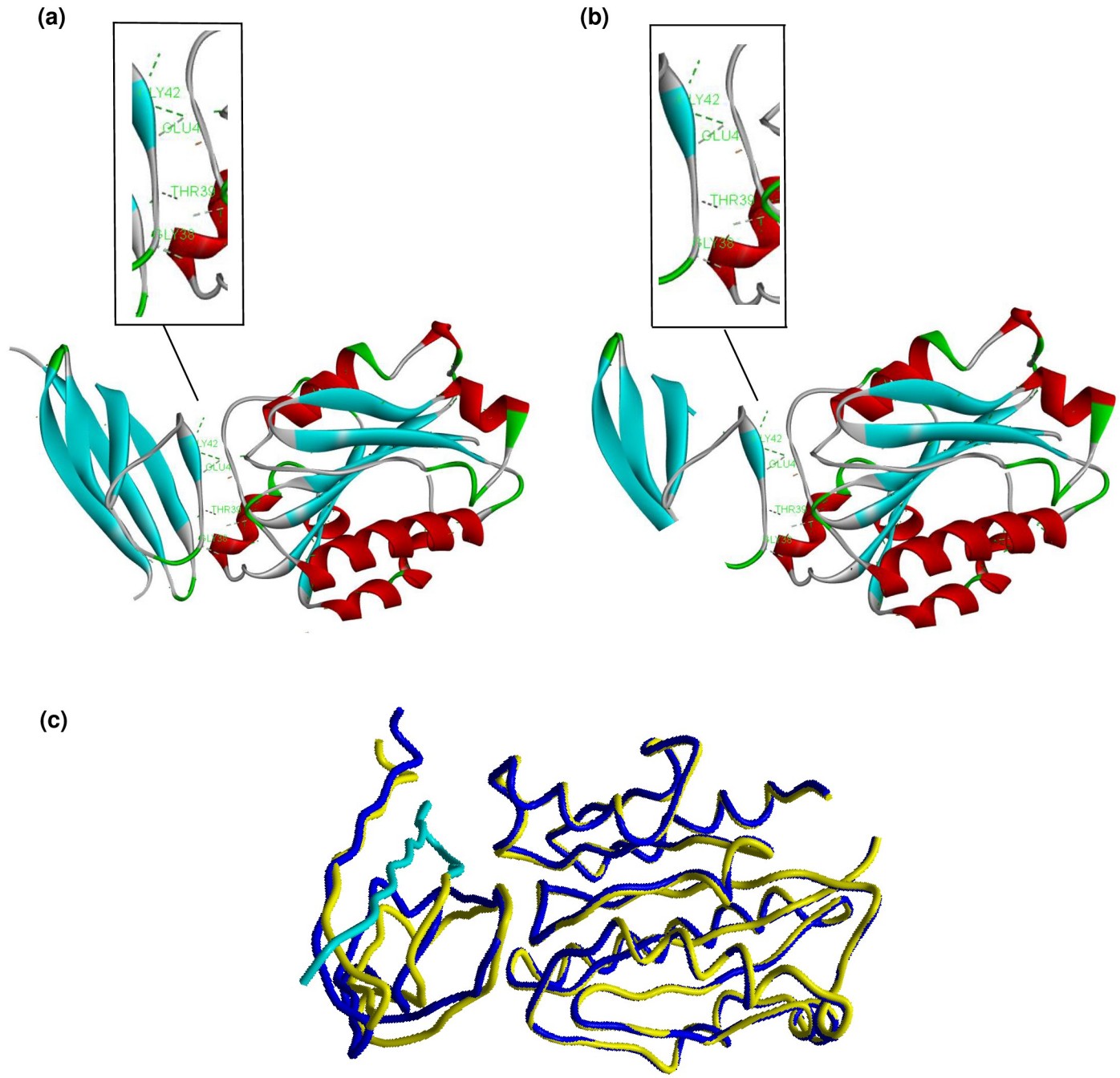

**Fig 5. The overall structure and protein-protein interactions of the ORF7a-LFA-1 complex for (a) wild ORF7a, (b) mutant ORF7a, and (c) LFA-1 superposition to wild ORF7a (Yellow) and mutant (Blue), with the truncated region highlighted in turquoise.**

TM score (0.70, TM-score > 0.5, indicating that the two proteins have the same fold). A flexible protein structure comparison between the model and template was achieved using FAT-CAT, where the obtained p-value (1.55e-15) and RMSD (0.06 Å) indicate that the two protein structures are significantly similar. After validation, quality check, adding polar hydrogen atoms, and adding Gasteiger charge of the selected model, it was then subjected to molecular

**Table 1. ORF7a-LFA-1 interaction score for both wild and mutant ORF7a protein.**

|  | Wild ORF7a | Mutant ORF7a |
|---|---|---|
| HADDOCK score | -67.0 +/- 7.4 | -63.6 +/- 3.2 |
| Cluster size | 7 | 11 |
| RMSD from the overall Lowest-energy structure | 0.9 +/- 0.5 | 1.4 +/- 1.0 |
| Van der Waals energy | -39.0 +/- 5.2 | -38.3 +/- 2.7 |
| Electrostatic energy | -157.4 +/- 9.9 | -148.5 +/- 8.7 |
| Desolvation energy | 0.9 +/- 1.8 | 0.9 +/- 2.5 |
| Restraints violation energy | 25.1 +/- 14.8 | 34.6 +/- 24.9 |
| Buried Surface Area | 1276.1 +/- 64.2 | 1248.8 +/- 61.9 |
| Z-Score | -2.1 | -1.7 |

docking alongside the wild ORF7a to assess the protein-protein interaction of ORF7a and LFA-1 I-domain (pdb: 7ci3). The latter is located on the cell membrane of human leukocytes, which is the target of ORF7a (Fig 5). The molecular docking results suggested that there was no considerable difference in the binding affinity of the mutant and wild ORF7a, and the HADDOCK scores were -67.0 and -63.6 for mutant and wild ORF7a, respectively (Table 1). The visualization of the active residue interactions showed that the functional amino acids Glu41, Gly42, Thr39. and Gly38 were contributing to hydrogen bonding with LFA-1 in both ORF7a structures (Fig 5), while the contribution of Ala66 was absent in the mutant structure.

## Phylogenetic tree

The phylogenetic tree confirmed the presence of different lineages belonging to multiple clusters (Fig 6). Seventy-six strains from the Iraqi population were distributed to nine different SARS-CoV-2 lineages, including B, B.1.1, B.1.177.21, B.1.621.1, B.1.1.374, B.1.351, B.1.1.7, B1.36, and B.1.438 corresponding to clades L, O, G, GH, GR, and GRY. The Iraqi strains of lineage B.1.1.7 were split into multiple sub-clusters most likely reflecting a large number of transmission chains in Iraq.

## Discussion

Complete-genome sequencing and phylogenetic analysis of SARS-CoV-2 strains is an essential approach for tracking the virus evolution and understanding the circulation of SARS-CoV-2 variants in Iraq. However, there is little genetic information about the SARS-CoV-2 outbreak in Iraq. Therefore, the current study aimed to provide some rudimentary information about the genotypes of SARS-CoV-2 that are circulating in the country.

In this analysis, 76 SARS-CoV-2 complete genomes were sequenced from Iraq. From these genomes, we identified nine lineages. Of these, four genome sequences accompanied four novel mutations (A2853T, T20780A, TG25774AT, and T26159A) that caused a change in the amino acids E45V (NSP3), M41K (NSP16), W128M (NS3), and V256D (NS3). As shown in resolved structures of proteins from related strains, the position at NSP16 that has the aa change M41K is incorporated in ligand binding and viral oligomerization interfaces. Also, the aa change W128M (NS3) is involved in the host cell protein/RNA interaction of SARS-CoV-2 [22], while the effect of the remaining mutations E45V (NSP3) and V256D (NS3) has not yet been investigated.

As expected, most (n = 73, 96%) of the sequences from the second epidemic wave in Iraq contained the amino acid replacement D614G in the spike proteins (Fig 3). D614G was the

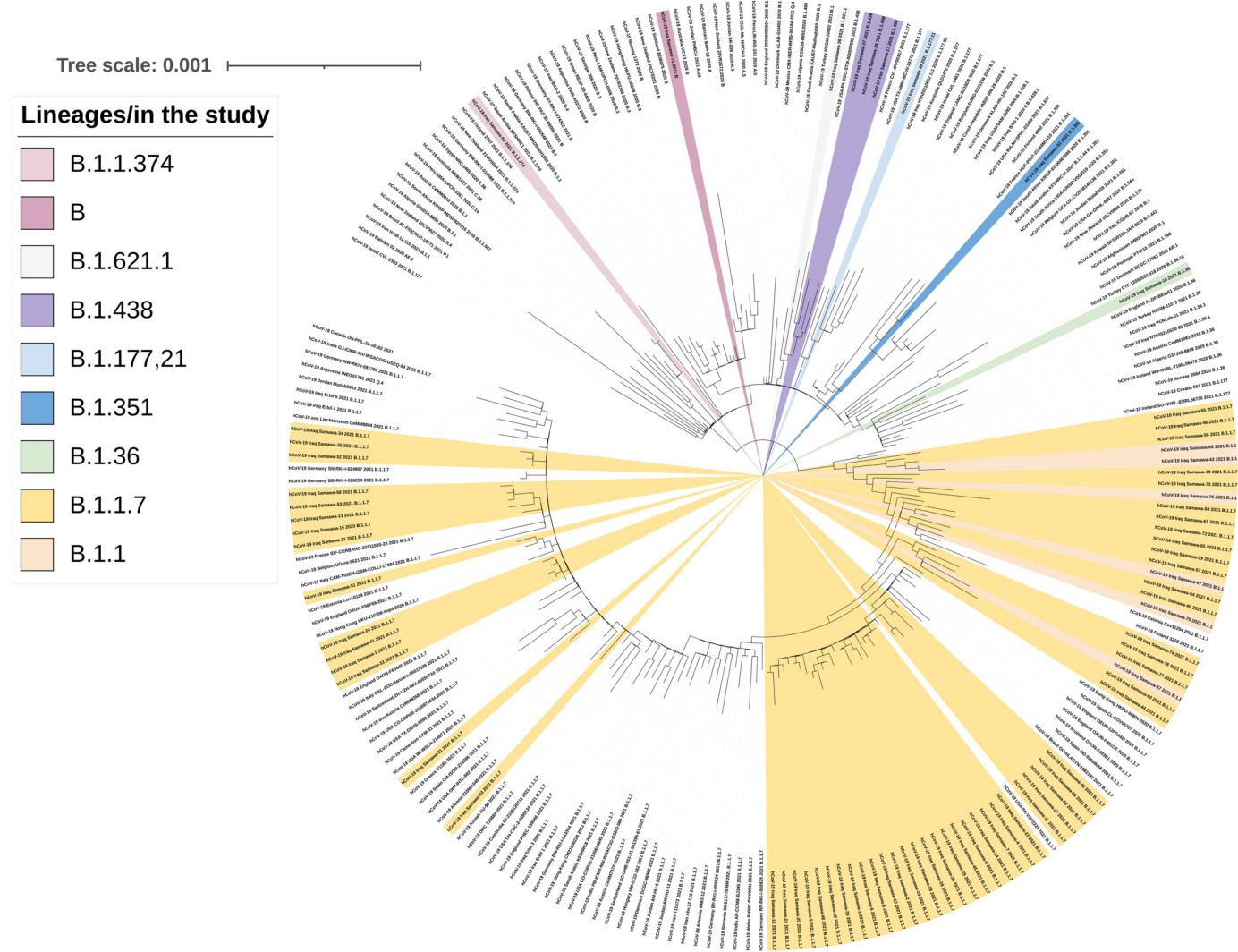

**Fig 6. Maximum-likelihood phylogenetic tree based on complete genomic sequences of SARS-CoV-2 constructed from 150 complete genome sequences from GISAID accessed 25 May 2021 and 76 sequences (yellow color) from Iraq.** B.1.1.7 was observed to split into GRY and GR and it was dominated by viruses as the largest clusters. Replicate number with 1,000 bootstrap on IQ-TREE.

first mutation in the spike glycoprotein that was first identified in Germany in January 2020 and became the dominant mutation in all the circulating strains worldwide by June 2020 [37].

The first SARS-CoV-2 genome sequence was reported on June 30, 2020, during the first wave occurrence in Iraq and belonged to the B.1/GH clade. According to the clade distribution, this clade diminished during the second wave. The B.1.1.7/GR and GRY clades were the most prevalent, which is consistent with the global distribution of SARS-CoV-2 clades in different countries.

Here, we report the first confirmed case of Alpha/B.1.1.7 variant of concern in Iraq (EPI_ISL_1524332), in a sample collected in December 27, 2020, followed by the recording of 60 cases (GISAID). Since then, the number of infected cases has risen to over seven thousand cases per day. This is likely due to the emergence of this variant, which is characterized by high transmissibility and pathogenicity.

The Beta variant (B.1.351) was first reported in South Africa in October 2020, and concerns about this variant are associated with high transmissibility, pathogenicity, and the limited protection of some vaccines against the infection [38]. Interestingly, we identified one strain belonging to B.1.351 for the first time in Iraq on February 26, 2021, which was collected from a patient without a history of travel, suggesting that this variant has been circulating locally before this date.

The Mu variant (B.1.621) has a set of interested mutations which raises the controversy about it has potential resistance to currently available COVID-19 vaccines (closer to what was raised about B.1.351) [39]. The Mu variant was detected in one Iraqi sample in February 2021, about a month after it was first emerged in Colombia.

Furthermore, we analyzed mutation profiles in the spike proteins of the alpha variant (B.1.1.7), showing that there is a genomic diversity of this variant in Iraq, which could be attributed to a variety of infection sources. It has been reported that the deletion 69–70 in the S protein causes a negative result from RT-PCR assays specific target for S-gene [40]. This specific deletion has occurred at high frequency in different countries and is currently geographically widespread. According to our results, this deletion was identified in 53 strains among 61 of the Alpha variant (Fig 4). In addition to this deletion, a cluster of aa mutations (Y144del, N501Y, A570D, D614G, P681H, T716I, S982A, and D1118H) were noticed in the spike proteins of the strains belonging to the Alpha variant (S2 Table). Fortunately, most of these aa changes are located outside the RBD; hence, they likely do not affect vaccine efficacy. On the other hand, a rare spike mutation (S605A) was detected in one strain (EPI_ISL_1524356). It has been reported that this mutation leads to the removal of a potential N-glycosylation site at position 603, where the motif at positions 603–605 changes from NTS (glycocylated) to NTA (non-glycocylated) and may also affect the antigenic properties of the virus "CoVsurver enabled by GISAID". Contrastingly, we detected a stop codon mutation (Q62*) at the ORF7a (NS7a) coding region (27577) in most of the Iraqi strains belonging to the Alpha variant, resulting in a truncated NS7a protein. The accessory protein ORF7 of SARS-CoV-2 is involved in modulation of host immune responses [36]. This motivated us to use an in silico approach to investigate the effect of Q62* on the function of NS7a using molecular docking scores (Table 1). The results predicted that NS7a was still able to bind to its target (LFA-1). Consistently, some strains with Q62* were derived from patients with severe infections, suggesting that truncated NS7a may not reduce the pathogenicity of the virus. Nevertheless, there is a possibility that the Q62 stop mutation affects ORF7a function, as Ongaro *et al.* computationally explored additional possible binding motifs in Orf7a for LFA-1, and they concluded in their study that the different binding modes of ORF7a cannot be ruled out [41]. Therefore, *in vitro* studies are required to explore the effect of the stop codon Q62* on the function of ORF7a.

The phylogenetic tree indicated that Iraqi B.1.1.7 strains form several subclusters, suggesting multiple introductions followed by local transmission, and most of the Iraqi strains clustered with the European strains, which may either reflect true importations or be due to unequal sampling efforts.

## Conclusions

The present study has a limited sample size, which may not represent the complete genomic diversity of SARS-CoV-2 in Iraq. It should be noted that the detected genomes may represent only locally distributed viral variations during the second epidemic wave, and the total diversity of SARS-CoV-2 circulating in Iraq is most likely significantly higher. Sequence analysis showed the transformation of the previously circulating strains from the first wave to the

dominance of Alpha variants that most likely surged during the second epidemic wave, as in most other countries. In addition, one Beta variant (B.1.351) was detected. Furthermore, we detected a prevalent NS7a Q62stop mutation among the Alpha variant strains in Iraq. In silico analysis suggested that there was no considerable difference in the binding affinity of mutant and wild NS7a to LFA-I, however, this mutation may have an effect on the binding affinity of NS7a towards another targets, especially there may be another targets that may interact directly or indirectly with NS7a, therefore more investigation is needed to explore the effect of Q62stop mutation on the function of NS7a protein.

## Supporting information

**S1 Table. Acknowledgement to the contributors of the SARS-CoV-2 sequences used in this study.**
(DOCX)

**S2 Table. A set of aa changes located in spike.**
(DOCX)

**S3 Table. GenBank accession numbers of SARS-CoV-2 strains analyzed in this study.**
(DOCX)

## Acknowledgments

We would like to thank Dr. Ryiad Abed-Ameer Halfi and Spec. Microbiologist Batool Kadham Salman, Ministry of Health, Iraq, and all the personnel from the unit of Molecular Virology, College of Medicine, Helsinki University, Finland, for their great efforts in this work.

## Author Contributions

**Conceptualization:** Nihad A. M. Al-Rashedi, Hussein Alburkat, Olli Vapalahti, Tarja Sironen, Teemu Smura.

**Data curation:** Nihad A. M. Al-Rashedi, Hussein Alburkat, Murad G. Munahi, Alaa Hameed, Olli Vapalahti, Tarja Sironen, Teemu Smura.

**Formal analysis:** Nihad A. M. Al-Rashedi, Murad G. Munahi.

**Funding acquisition:** Nihad A. M. Al-Rashedi.

**Methodology:** Nihad A. M. Al-Rashedi, Hussein Alburkat, Abas O. Hadi, Murad G. Munahi, Ali Jasim, Alaa Hameed, Olli Vapalahti, Tarja Sironen, Teemu Smura.

**Project administration:** Nihad A. M. Al-Rashedi.

**Resources:** Basel Saber Oda, Kareem Moamin Lilo.

**Software:** Nihad A. M. Al-Rashedi, Murad G. Munahi, Teemu Smura.

**Supervision:** Nihad A. M. Al-Rashedi.

**Writing – original draft:** Nihad A. M. Al-Rashedi, Murad G. Munahi, Alaa Hameed, Basel Saber Oda, Kareem Moamin Lilo, Laith A. H. AlObaidi.

**Writing – review & editing:** Hussein Alburkat, Olli Vapalahti, Tarja Sironen, Teemu Smura.

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
