## [Decision Letter · Decision Letter 0]

11 Jan 2022

PONE-D-21-36713High prevalence of an alpha variant lineage with a premature stop codon in ORF7a in Iraq, winter 2020-2021PLOS ONE

Dear Dr. Al-Rashedi,

Thank you for submitting your manuscript to PLOS ONE. After careful consideration, we feel that it has merit but does not fully meet PLOS ONE’s publication criteria as it currently stands. Therefore, we invite you to submit a revised version of the manuscript that addresses the points raised during the review process.

We look forward to receiving your revised manuscript.

Kind regards,

Junwen Wang, Ph.D.

Academic Editor

PLOS ONE

Journal Requirements:

Reviewers' comments:

Reviewer's Responses to Questions

**Comments to the Author**

1. Is the manuscript technically sound, and do the data support the conclusions?

Reviewer #1: Partly

Reviewer #2: Partly

2. Has the statistical analysis been performed appropriately and rigorously? 

Reviewer #1: N/A

Reviewer #2: I Don't Know

3. Have the authors made all data underlying the findings in their manuscript fully available?

Reviewer #1: Yes

Reviewer #2: Yes

4. Is the manuscript presented in an intelligible fashion and written in standard English?

Reviewer #1: Yes

Reviewer #2: Yes

5. Review Comments to the Author

Reviewer #1: The manuscript represents a welcome contribution to the analysis of circulating SARS-CoV-2 variants and strains. The laboratory and analytical tools are adequate and the description of the work is, on the whole, clear. The major drawback is the limited number of sequences generated and analyzed which cannot be representative to the actual population of strains circulating at the time in the country. There is no information about the selection criteria for the samples that were analyzed either; some of the comments in the discussion chapter as well as in conclusions are therefore overstated. The manuscript offers only a limited view of the circulating sequences in Iraq at the time; the text should be correspondingly changed to reflect this.

Specific comments:

Lines 33-35: analysis of only 76 genomes cannot contribute to understanding the molecular epidemiology of the second COVID-19 wave in Iraq.

Line 54: there is not enough data to support the statement that the developing countries have been selectively affected

Line 67: there is not enough evidence of increased severity of disease in VOCs.

Sampling: what are the selection criteria for the 76 samples? A “high copy number” is mentioned; what was the threshold Ct in the particular RT-PCR setting?

Fig 6: the distribution of lineages in the tree is not clearly enough represented and colored

Lines 304-306: possible consequences of the novel mutations should be further discussed.

Lines 368-369: only a very limited sample of genomic sequences of SARS-CoV-2 from Iraq are actually reported.

Reviewer #2: Observations:

-On line 42, the authors state that there is a high frequency of this stop codon mutation (88%). However, they also state (line 239) that this mutation occurred in 26 strains as in Supplementary table S3. Therefore, the authors need to either explain the difference or to modify accordingly on this line to: 34.21% if related to the all 76 sequenced strains, or to 43.33% if related to the 60 B.1.1.7 strains only.

-On line 130, the authors state that the NovaSeq system produced 1.222.270 reads – I guess this is the average per sample? Otherwise, if compared to the 76 sequenced samples, the average would be about 16k reads per sample, which is very low. Novaseq generates plenty of data if there’s nothing wrong with the samples to be sequenced.

-On lines 132 to 133, there should be “assembled”, “called” and “called”.

-On line 211, the authors state that there are 33 strains that belong to GRY and 29 that belong to GR. In the Supplementary table S2 there are 4 other strains that belong to GH and 1 to O clade, therefore totalling 67 strains. The authors should also mention what happens with the other 9 sequenced strains (up to 76) – do they have any mutation; to what clades they belong to?

-On lines 228 to 231, the authors suddenly mention a rare S605A mutation, without prior description. They could discuss on the possible effects that this mutation produces (if any): where is located, what happens or would happen with the local conformation of the Spike, etc. Is there a good coverage on this mutation? Also, there’s a typo – it should be “the S605A mutation IN THE spike gene”.

-Similarly, on lines 233-234, a short discussion would be advised for the other 4 novel mutations. Do these mutations have a high sequencing coverage?

-On lines 251 to 256, the authors mention a paper about the possible binding potential of ORF7a to LFA-1 and MAC-1, based on docking studies. They also discuss about some aminoacids that are important for the function of ORF7a. They don’t tell if those aminoacids fold in close proximity in 3D. They also don’t state what happens with the conformation locally, since there are truncated aminoacids due to Q62stop codon.

-On lines 267 to 272, the authors forgot to mention the PDB code for the LFA-1 3D structure, used in the docking simulation. They say that there’s no significant difference in the binding affinity but would’ve been useful to add some details on the docking simulation (how many docking conformations were generated and how do they cluster, and if there are multiple clusters, what were the ΔGs for each cluster). For instance, another study by Ongaro et al., JCIM, May 2021, suggests more than a possible binding motif between Orf7a and LFA-1, as well as the conformational similarity between ORF7a and ICAM3. So, it would be useful to share the whole image of the docking simulation.

-On line 274, in Figure 5 the authors show only two images, side by side, of the WT and mutant ORF7a. The viewing angle is not the same – one structure is rotated with respect to the other. It would’ve been better to represent the two structures superimposed and represented not as a solvent-accessible surface but rather as cartoon or CA-trace ribbon, in order to see the secondary structures in the protein. Also, a zoom-in into the region of Q62stop codon (in order to see the local changes) would be more relevant, and maybe represent as side-chain some of the interface residues (from each protein) involved in the interaction, especially near Q62.

-On line 290, there should be “sequences” instead of “sequencing”.

-On lines 311 to 317, starting with “Despite the D614G….” – there is redundant information that can be removed.

-On lines 346 to 348, the Supplementary table S3, to which the authors refer, contains epidemiological info, and underlines only the stop codon mutation, without any connection to the other mutations discussed in the phrase.

-On lines 374 to 375, the authors suggest that there is no significant difference in the binding affinity. I would underline the fact that authors consider only this LFA-1 possible interactor, but neglect to address other interactors that may interact directly or indirectly (through a network of interactors) with ORF7a, as can be found when searching for similar data in interactomics databases such as BioGrid.

-The phylogenetic tree figure could be improved: it’s hard to see in the chosen representation how the strains split/cluster; there is no highlight on the clades or on lineages; no outgroup mention.

6. PLOS authors have the option to publish the peer review history of their article (what does this mean?). If published, this will include your full peer review and any attached files.

Reviewer #1: No

Reviewer #2: No

---

## [Author Response · Author response to Decision Letter 0]

23 Feb 2022

Response to Decision Letter and Reviewer Comments 

Manuscript number: PONE-D-21-36713

Title: High prevalence of an alpha variant lineage with a premature stop codon in ORF7a in Iraq, winter 2020-2021

Reviewers' Comments:

Reviewer #1

The manuscript represents a welcome contribution to the analysis of circulating SARS-CoV-2 variants and strains. The laboratory and analytical tools are adequate and the description of the work is overall, clear. The major drawback is the limited number of sequences generated and analyzed which cannot be representative to the actual population of strains circulating at the time in the country. There is no information about the selection criteria for the samples that were analyzed either; some of the comments in the discussion chapter as well as in conclusions are therefore overstated. The manuscript offers only a limited view of the circulating sequences in Iraq at the time; the text should be correspondingly changed to reflect this.

Specific comments:

Lines 33-35: analysis of only 76 genomes cannot contribute to understanding the molecular epidemiology of the second COVID-19 wave in Iraq.

We agree that this number of genome sequences is not enough to cover the COVID-19 molecular epidemiology of the whole country.

However, we think this study can give some insight on the strains circulated during the second wave of COVID-19 in Iraq although its pilot study. We have now addressed the limitations of the study is more clearly in lines 382-386.

Line 54: there is not enough data to support the statement that the developing countries have been selectively affected

We agree, and revised this phrase and the reference [3] to:

 ''affecting humans worldwide. [3]

[3 ] Deng G, Shi J, Li Y, Liao Y. The COVID‐19 pandemic: shocks to human capital and policy responses. Account Finance. 2021;61(4):5613-5630. https://doi.org/10.1111/acfi.12770

Line 67: there is not enough evidence of increased severity of disease in VOCs.

According to your recommendations, the phrase has been modified to mean that all SARS-CoV-2 VOCs are associated with an increase in hospitalization, including ICU, and deaths, but there is a different degree of each one compared to the wild-type strains. There is no strong evidence to say that SARS-CoV-2 VOCs cause the severity of the disease, but we can say they have a significant impact on the risk of hospitalization, ICU admissions, and deaths (7).

Other analysis suggests that B.1.1.7 is not only more transmissible than pre-existing SARS-CoV-2 variants, but may also cause more severe illness, that the study depicted the 1,61 x higher case fatality ratio of alpha (B.1.1.7) as compared to "wild type" in the UK (8).

7- Lin L, Liu Y, Tang X, He D. The Disease Severity and Clinical Outcomes of the SARS-CoV-2 Variants of Concern. Front. Public Health. 2021;9:775224. doi: 10.3389/fpubh.2021.775224

8- Davies, N.G., Jarvis, C.I., CMMID COVID-19 Working Group. et al. Increased mortality in community-tested cases of SARS-CoV-2 lineage B.1.1.7. Nature 593, 270–274 (2021). https://doi.org/10.1038/s41586-021-03426-1

Sampling: what are the selection criteria for the 76 samples? A “high copy number” is mentioned; what was the threshold Ct in the particular RT-PCR setting?

We have now included this information to the manuscript. The threshold CT value <24 was used for all samples.

Fig 6: the distribution of lineages in the tree is not clearly enough represented and colored

In Figure 6, the lineages of strains have been added.

Lines 304-306: possible consequences of the novel mutations should be further discussed.

According to your recommendation, we have added a short discussion about the effect (including ligand binding, viral oligomerization and host cell protein/RNA interactions) of these mutations in the discussion section (lines 314-319). However, for some of the mutations, the phenotypic effect is not known or is difficult to predict.

Lines 368-369: only a very limited sample of genomic sequences of SARS-CoV-2 from Iraq are actually reported

According to your recommendation, we have stated a limitation of the study at the start of the conclusion:

“The present study has a limited sample size, which may not represent the complete genomic diversity of SARS-CoV-2 in Iraq. It should be noted that the detected genomes may represent only locally distributed viral variations during the second epidemic wave, and the total diversity of SARS-CoV-2 circulating in Iraq is most likely significantly higher”. 

Reviewer #2

-On line 42, the authors state that there is a high frequency of this stop codon mutation (88%). However, they also state (line 239) that this mutation occurred in 26 strains as in Supplementary table S3. Therefore, the authors need to either explain the difference or to modify accordingly on this line to: 34.21% if related to the all 76 sequenced strains, or to 43.33% if related to the 60 B.1.1.7 strains only.

Thank you very much for your correction. This was a typo, and the percentage is corrected (43.33%).

-On line 130, the authors state that the NovaSeq system produced 1.222.270 reads – I guess this is the average per sample? Otherwise, if compared to the 76 sequenced samples, the average would be about 16k reads per sample, which is very low. Novaseq generates plenty of data if there’s nothing wrong with the samples to be sequenced.

This is added to the text ‘a range of 1.4 to 3.3 million paired-end sequence reads per sample.’

-On lines 132 to 133, there should be “assembled”, “called” and “called”.

These phrases have been changed accordingly.

-On line 211, the authors state that there are 33 strains that belong to GRY and 29 that belong to GR. In the Supplementary table S2 there are 4 other strains that belong to GH and 1 to O clade, therefore totalling 67 strains. The authors should also mention what happens with the other 9 sequenced strains (up to 76) – do they have any mutation; to what clades they belong to?

We appreciate for pointing out this inconsistency. The missing 9 sequences were added to the supplementary table S2 and distributed according to mutations and lineages as follows (GR, n=6), (O, n=1), (L, n=1) and (G, n=1).

On line 211, we reported only the clades that carry the most common cluster of spike protein mutations “H69del, V70del, Y144del, A570D, D614G, P681H, T716I, S982A, and D1118H”, where the remaining clades (G, GH, O, and L) are not involved since they don’t carry this mutation.

-On lines 228 to 231, the authors suddenly mention a rare S605A mutation, without prior description. They could discuss on the possible effects that this mutation produces (if any): where is located, what happens or would happen with the local conformation of the Spike, etc. Is there a good coverage on this mutation? Also, there’s a typo – it should be “the S605A mutation IN THE spike gene”.

Thank you for pointing this out. We have now revised the corresponding text and added discussion on lines 357-364. Also, the typo is corrected in the text. 

-Similarly, on lines 233-234, a short discussion would be advised for the other 4 novel mutations. Do these mutations have a high sequencing coverage?

We have added a short discussion about the effects of these mutations in the discussion section (lines 313–318). The sequencing coverage of these 4 mutations has a range from 80553 to 14774x.

-On lines 251 to 256, the authors mention a paper about the possible binding potential of ORF7a to LFA-1 and MAC-1, based on docking studies. They also discuss about some amino acids that are important for the function of ORF7a. They don’t tell if those amino acids fold in close proximity in 3D. They also don’t state what happens with the conformation locally, since there are truncated amino acids due to Q62stop codon.

We agree with the reviewer’s assessment, accordingly, the corresponding text has been revised and the following phrase added lines 256-258:

“The truncated amino acids are mainly located in one β sheet strand (βG) in the ectodomain. Only four amino acids are located in the β sheet strand βf (Fig 5).”

Fig 5 has replaced by a new one to visualize the functional amino acids, as well as the following phrase added lines 275-279 and the caption of Figure 5: lines 281-283:

“The visualization of the active residue interactions showed that the functional amino acids Glu41, Gly42, Thr39. and Gly38 were contributing to hydrogen bonding with LFA-1 in both ORF7a structures (Fig 5), while the contribution of Ala66 was absent in the mutant structure.”

-On lines 267 to 272, the authors forgot to mention the PDB code for the LFA-1 3D structure, used in the docking simulation. They say that there is no significant difference in the binding affinity but would have been useful to add some details on the docking simulation (how many docking conformations were generated and how do they cluster, and if there are multiple clusters, what were the ΔGs for each cluster). For instance, another study by Ongaro et al., JCIM, May 2021, suggests more than a possible binding motif between Orf7a and LFA-1, as well as the conformational similarity between ORF7a and ICAM3. So, it would be useful to share the whole image of the docking simulation.

We have added the pdb code   of LFA-1 (PDB: 3f78) as recommended (lines 265-266).

It seems that we used the word "significant" incorrectly, where we didn’t mean statistically significant. All we wanted to say is that the difference in the binding affinity between both systems (wild and mutant) is not large (quantitatively not statistically), so to remove the confusion, we replaced the word "significant" with "considerable" in line 273.

In their study, Ongaro et al reported the ΔGs and clustering of the docking results. Because they were searching for a binding motif, it seemed like a blind docking, so we performed only one docking simulation for each ORF7a structure (wild and mutant).

-On line 274, in Figure 5 the authors show only two images, side-by-side, of the WT and mutant ORF7a. The viewing angle is not the same – one structure is rotated with respect to the other. It would’ve been better to represent the two structures superimposed and represented not as a solvent-accessible surface but rather as cartoon or CA-trace ribbon, in order to see the secondary structures in the protein. Also, a zoom-in into the region of Q62stop codon (in order to see the local changes) would be more relevant, and maybe represent as side-chain some of the interface residues (from each protein) involved in the interaction, especially near Q62.

In Figure 5, the two structures have been superimposed and represented by Ribbon and the truncated region has been highlighted. The overall structures and protein-protein interactions of the ORF7a-LFA-1 complex for the wild ORF7a and mutant ORF7a as well as a zoomed-in view of the interaction region have been presented.

-On line 290, there should be “sequences” instead of “sequencing”.

Revised accordingly.

-On lines 311 to 317, starting with “Despite the D614G….” – there is redundant information that can be removed.

This has been removed as recommended.

-On lines 346 to 348, the Supplementary table S3, to which the authors refer, contains epidemiological info, and underlines only the stop codon mutation, without any connection to the other mutations discussed in the phrase.

Thanks for the comment. It is a typo, since the intended table is S2 instead of S3, where table S2 contains the corresponding mutations. So it has been corrected to “Supplementary table S2”.

-On lines 374 to 375, the authors suggest that there is no significant difference in the binding affinity. I would underline the fact that authors consider only this LFA-1 possible interactor, but neglect to address other interactors that may interact directly or indirectly (through a network of interactors) with ORF7a, as can be found when searching for similar data in interactomics databases such as BioGrid.

Thank you for notifying us that there may be another possible interactor. The following phrase has been added to the corresponding text on lines 369-373 and reference 40 “Nonetheless, there is a possibility that the Q62 stop mutation affects ORF7a function, as Ongaro et al. computationally explored additional possible binding motifs in Orf7a for LFA-1.They concluded in their study that the different binding modes of ORF7a can not be ruled out [40]. Therefore, in vitro studies are required to explore the effect of the stop codon Q62* on the function of ORF7a.”

-The phylogenetic tree figure could be improved: it’s hard to see in the chosen representation how the strains split/cluster; there is no highlight on the clades or on lineages; no outgroup mention.

In Figure 6, the lineages of strains have been added..

---

## [Decision Letter · Decision Letter 1]

17 Mar 2022

PONE-D-21-36713R1High prevalence of an alpha variant lineage with a premature stop codon in ORF7a in Iraq, winter 2020-2021PLOS ONE

Dear Dr.Nihad Al-Rashedi,

Thank you for submitting your manuscript to PLOS ONE. After careful consideration, we feel that it has merit but does not fully meet PLOS ONE’s publication criteria as it currently stands. Therefore, we invite you to submit a revised version of the manuscript that addresses the points raised during the review process.

We look forward to receiving your revised manuscript.

Kind regards,

Muhammad Qasim, Ph.D

Academic Editor

PLOS ONE

Additional Editor Comments:

The sample size is not statistically significant. At least 500 sample needs to be analyzed before inferring something.

Patients were selected just on the basis of viral titre (a high copy number of the virus (Ct values

110 <24)), the questions is were these patients immune competent before they caught infection? Do these patients have no comorbidity? The manuscript lacks information about clinical details of subjects.

Those who died, did have any other disease that after infection might have resulted in death? Because the major portion of enrolled patients had mild to moderate symptoms in spite of high copy number of the virus.

Reviewers' comments:

Reviewer's Responses to Questions

**Comments to the Author**

1. If the authors have adequately addressed your comments raised in a previous round of review and you feel that this manuscript is now acceptable for publication, you may indicate that here to bypass the “Comments to the Author” section, enter your conflict of interest statement in the “Confidential to Editor” section, and submit your "Accept" recommendation.

Reviewer #1: All comments have been addressed

Reviewer #2: All comments have been addressed

2. Is the manuscript technically sound, and do the data support the conclusions?

Reviewer #1: Yes

Reviewer #2: Yes

3. Has the statistical analysis been performed appropriately and rigorously? 

Reviewer #1: N/A

Reviewer #2: I Don't Know

4. Have the authors made all data underlying the findings in their manuscript fully available?

Reviewer #1: Yes

Reviewer #2: Yes

5. Is the manuscript presented in an intelligible fashion and written in standard English?

Reviewer #1: Yes

Reviewer #2: Yes

6. Review Comments to the Author

Reviewer #1: (No Response)

Reviewer #2: (No Response)

7. PLOS authors have the option to publish the peer review history of their article (what does this mean?). If published, this will include your full peer review and any attached files.

Reviewer #1: No

Reviewer #2: No

---

## [Author Response · Author response to Decision Letter 1]

29 Mar 2022

Dear Editor

Thank you very much for your valuable review. Here is a point-by-point response to the editor's comments.

As mentioned in response to the first and second reviewers, we agree that this number of genome sequences is not enough to cover the COVID-19 molecular epidemiology of the entire country. However, due to the limited sequencing capacity, there is currently very little information on the SARS-CoV-2 variants circulating in Iraq. Given that the global availability of SARS-CoV-2 genomic surveillance is extremely biased, we believe that covering this gap in mid-and low-income countries is highly valuable, and, albeit imperfect, the current study provides insight into the strains circulated during the second wave of COVID-19 in Iraq.

We have now more clearly addressed the limitations of the study in lines 382–386. 

"The present study has a limited sample size, which may not represent the complete genomic diversity of SARS-CoV-2 in Iraq. It should be noted that the detected genomes may represent only locally distributed viral variations during the second epidemic wave, and the total diversity of SARS-CoV-2 circulating in Iraq is most likely significantly higher".

Regarding clinical details, we have now added information on comorbidities to Table S3. As mentioned earlier, the sample size is too small to correlate any variables with the disease outcome or clinical manifestations, and therefore, we are reluctant to speculate on these in the manuscript. Instead, we focus on genomic characterization and in silico assessment of the potential phenotypic effects of the observed amino acid substitutions.

---

## [Decision Letter · Decision Letter 2]

6 Apr 2022

High prevalence of an alpha variant lineage with a premature stop codon in ORF7a in Iraq, winter 2020-2021

PONE-D-21-36713R2

Dear Dr. Nihad Al-Rashedi,

We’re pleased to inform you that your manuscript has been judged scientifically suitable for publication and will be formally accepted for publication once it meets all outstanding technical requirements.

Kind regards,

Wenping Gong, Ph.D.

Academic Editor

PLOS ONE

Additional Editor Comments (optional):

Reviewers' comments:

Reviewer's Responses to Questions

**Comments to the Author**

1. If the authors have adequately addressed your comments raised in a previous round of review and you feel that this manuscript is now acceptable for publication, you may indicate that here to bypass the “Comments to the Author” section, enter your conflict of interest statement in the “Confidential to Editor” section, and submit your "Accept" recommendation.

Reviewer #1: All comments have been addressed

Reviewer #2: All comments have been addressed

2. Is the manuscript technically sound, and do the data support the conclusions?

Reviewer #1: (No Response)

Reviewer #2: Yes

3. Has the statistical analysis been performed appropriately and rigorously? 

Reviewer #1: (No Response)

Reviewer #2: N/A

4. Have the authors made all data underlying the findings in their manuscript fully available?

Reviewer #1: (No Response)

Reviewer #2: Yes

5. Is the manuscript presented in an intelligible fashion and written in standard English?

Reviewer #1: (No Response)

Reviewer #2: Yes

6. Review Comments to the Author

Reviewer #1: (No Response)

Reviewer #2: (No Response)

7. PLOS authors have the option to publish the peer review history of their article (what does this mean?). If published, this will include your full peer review and any attached files.

Reviewer #1: No

Reviewer #2: No

---

## [Editor Report · Acceptance letter]

17 May 2022

PONE-D-21-36713R2 

High prevalence of an alpha variant lineage with a premature stop codon in ORF7a in Iraq, winter 2020-2021 

Dear Dr. Al-Rashedi:

I'm pleased to inform you that your manuscript has been deemed suitable for publication in PLOS ONE. Congratulations! Your manuscript is now with our production department. 

Kind regards, 

on behalf of

Dr. Wenping Gong 

Academic Editor

PLOS ONE